# Simultaneous detection of *Mycobacterium tuberculosis* complex and resistance to Rifampicin and Isoniazid by MDR/MTB ELITe MGB® Kit for the diagnosis of tuberculosis

**Francesco Bisognin, Giulia Lombardi** **\*, Chiara Finelli, Maria Carla Re, Paola Dal Monte**

Microbiology Unit—Department of Experimental, Diagnostic and Specialty Medicine, S. Orsola-Malpighi University Hospital, Bologna, Italy

\* g.lombardi@unibo.it

**Data Availability Statement:** All de-identified data used in this study are available from the "Database Bisognin F et al. ELITe" file held in the Figshare

## Abstract

The MDR/MTB ELITe MGB® Kit on the ELITe InGenius® platform (ELITechGroup SpA, Italy) is the first system for simultaneous detection of the *Mycobacterium tuberculosis* complex (MTBc) genome and the main mutations responsible for resistance to Isoniazid (*inhA*, *katG*) and Rifampicin (*rpoB*), from decontaminated and heat inactivated samples. In this study we compared the performance of the MDR/MTB ELITe MGB® Kit (ELITe) with culture in 100 pulmonary and 160 extra-pulmonary samples. The sensitivity and specificity of ELITe compared to culture for pulmonary samples were 98.0% and 98.0% respectively; for extra-pulmonary samples the overall sensitivity was 86.3% (80% for urine, 85% for biopsy and gastric aspirate and 95% for cavitary fluid) and specificity was 100%. Genotypic Isoniazid and Rifampicin susceptibility typing was feasible in 96% of sputum MTBc-positive samples and 43% of extra-pulmonary samples; all samples were found to be drug susceptible by phenotypic and ELITe (100% agreement). Detection of mutations in the *rpoB, kat G* or *inhA* genes was evaluated on 300 spiked samples (60 per biological matrix) and all resistance profiles were correctly identified by ELITe. Molecular agreement between ELITe and Xpert was 98.0% and 93.3% for pulmonary and extra-pulmonary samples, respectively. In conclusion, our results provide evidence to support the use of MDR/MTB ELITe MGB® Kit in combination with ELITe InGenius® for the diagnosis of MTBc and the detection of Rifampicin and Isoniazid resistance-related mutations in both pulmonary and extra-pulmonary samples. This system simplifies the laboratory workflow, shortens report time and is an aid in choosing appropriate therapeutic treatment and patient management.

## Introduction

According to the last WHO Tuberculosis (TB) report, in 2018 an estimated 10.0 million (range, 9.0–11.1 million) people fell ill with TB globally and an estimated 280,000 new and relapse TB cases occurred in the WHO European region. In addition, there were 77,000

public repository at the following URL: https://doi.org/10.6084/m9.figshare.12123585.v1

**Funding:** The authors received no specific funding for this work.

**Competing interests:** The authors have declared that no competing interests exist.

estimated cases of multidrug-resistant (MDR) and rifampicin-resistant (RR) TB among notified pulmonary cases in the European region and the global average of isoniazid resistance without concurrent rifampicin resistance was 7.2% in new TB cases and 11.6% in previously treated TB cases [1].

The onset of this resistance is caused by incomplete treatment and/or inadequate therapy [2]. In fact, all patients initially receive standard daily anti-TB treatment based on preliminary results, which is then tailored accordingly as soon as drug susceptibility test results are available. [3]

*Mycobacterium tuberculosis* complex (MTBc) includes the species *M. tuberculosis*, *M. africanum*, *M. bovis*, *M. microti* and *M. canettii*, that cause tuberculosis disease, with almost identical nucleotide sequences and totally identical 16S rRNA sequences [4], The identification and drug susceptibility results *of* MTBc can require several weeks in smear-negative samples, and consequently optimal treatment may be delayed. In addition, according to the last ECDC Report [5], in European countries 17% of incident TB cases were extra-pulmonary (EPTB). EPTB is characterized by a very low bacterial load and remains undiagnosed for a long time in a considerable number of cases due to atypical presentation, often simulating neoplasia and/or inflammatory disorders [6].

A great deal of effort is currently focused on developing rapid and reliable molecular diagnosis of drug-resistant TB in order to initiate correct therapy [7].

MDR/MTB ELITe MGB® Kit (ELITechGroup, Italy) is a new multiplex, ultra-sensitive, real time PCR assay (limit of detection 6 CFU/mL) used for the detection of MTBc DNA as well as Rifampicin and Isoniazid resistance [8]. The assay workflow of the ELITe InGenius® system integrates the extraction and purification of nucleic acids, real-time PCR amplification, detection of the target sequence with melt-curve capability and result interpretation.

In this retrospective study, we assessed the performance of the MDR/MTB ELITe MGB® Kit (ELITe) on pulmonary and extra-pulmonary specimens in comparison with culture as well as its ability to detect Rifampicin and Isoniazid resistance on different specimen matrices spiked with three drug-resistant strains. Agreement with Xpert MTB/RIF(Cepheid, USA) was also evaluated.

## Materials and methods

### Study design

This is a retrospective study performed on frozen (-20˚C) pulmonary and extra-pulmonary samples, collected between January 2017 and June 2018, and previously processed for MTBc detection by smear, culture and Xpert, at the Microbiology Unit of the S. Orsola-Malpighi Hospital, Bologna (Italy).

First, the sensitivity of the MDR/MTB ELITe MGB® Kit (ELITe) was assessed on 50 sputum and 80 extra-pulmonary samples (20 urine, 20 biopsy, 20 cavitary fluid, 20 gastric aspirate) which were culture-positive for drug-susceptible MTBc. Specificity was evaluated on 50 MTBc culture-negative sputum samples and 80 MTBc culture-negative extra-pulmonary samples.

Secondly, detection of mutations in the *rpoB*, *katG* or *inhA* genes in different sample matrices was assessed. Previously processed MTBc culture-negative specimens were spiked with 3 MTBc isolates which were phenotypically resistant to Rifampicin and/or Isoniazid, carrying mutations in the *rpoB*, *katG* or *inhA* genes. In order to have a sufficient number of resistant samples for this diagnostic validation, 20 samples of each biological matrix were spiked with each of the 3 mutated MTBc strains, making a total of 300 spiked samples.

All frozen samples were anonymized and heat-inactivated for analysis with the MDR/MTB ELITe MGB® Kit on ELITe InGenius® system. The study was approved by the Ethics

Committee of Area Vasta Emilia Centro (AVEC), Bologna, Italy (Study protocol n.137/2017/U/Tess). Informed consent was not required as the data were analysed anonymously.

## Samples processing

Pre-treatment depended on the type of sample. Pulmonary and gastric aspirate samples were fluidified, if necessary, with Sputasol solution [9]. Urine and cavitary fluids (pleural, abdominal and ascitic fluids) were centrifuged and the supernatant removed to leave a final volume of 5 mL; urine pellets were also washed with 0.9% saline solution. Biopsies were mechanically homogenized with the addition of 0.9% saline solution to reach a volume of 5 mL. All specimens were digested and decontaminated using BBL MycoPrep™ solution (Becton Dickinson, USA) according to the manufacturer's instructions, and re-suspended in 2.5 mL of phosphate buffered solution [10]. Two mL were used in the routine work-flow for MTBc detection by acid-fast microscopy (Ziehl-Neelsen stain), culture in solid media (Lowenstein-Jensen, Heipha Diagnostics, Germany),culture in liquid media (MGIT 960, Becton Dickinson), and Xpert MTB/RIF on GeneXpert platform (Xpert, Cepheid, USA). For this validation study 0.5 mL were stored at -20˚C.

Phenotypic Antimicrobial Susceptibility Testing (AST) was performed by the "gold standard" automatic MGIT 960 System (Becton Dickinson, USA) on all MTBc-positive cultures. GenoType MTBDRplus VER 2.0 (Hain Lifescience, Germany) was performed on MTBc strains phenotypically resistant to Rifampicin and/or Isoniazid to detect mutations in the *rpoB*, *katG* or *inhA* genes.

The complete microbiological characterization of the pulmonary and extra-pulmonary samples is shown in Table 1.

## Sample inactivation

0.5 mL of each decontaminated sample were defrosted and inactivated at 95˚C for 30 minutes in a dry block (ThermoStat plus, Eppendorf) where the temperature was checked manually with a thermometer. To verify MTBc inactivation, 250 μL of all specimens (n = 560) were cultured in solid media and incubated at 37˚C for 8 weeks [1]. The remaining 250 μL were frozen (-20˚C) for analysis with MDR/MTB ELITe MGB® Kit on ELITe InGenius® system.

## MDR/MTB ELITe MGB® Kit on ELITe InGenius® system

200 μL of decontaminated and heat-inactivated sputum (n = 160) and extra-pulmonary samples (n = 400) were analysed by ELITe according to the manufacturer's instructions [8].

If the MTBc concentration is >2,500 CFU/mL, corresponding to MTB (IS6110 probe) Ct ≤31, melting temperature (Tm) analysis is automatic. When Ct is > 31, the sample is reported

**Table 1. Microbiological characteristics of samples included in the study.**

| Sample type | n. of samples tested | AFB Microscopy positive | Xpert MTB/RIF positive | MTBc Culture positive |
|---|---|---|---|---|
| Sputum | 160 | 44 | 50 | 50 |
| Urine | 100 | 2 | 8 | 20* |
| Biopsy | 100 | 4 | 20 | 20 |
| Cavitary Fluid | 100 | 4 | 18 | 20 |
| Gastric Aspirate | 100 | 0 | 18 | 20 |

*8 of the 20 MTBc positive urine samples were spiked at 20 CFU/ml with MTBc sensitive Strain 1.

AFB: Acid-fast bacilli.

as "MTB DNA Detected, typing not feasible" and the Tm of the target probes (*rpo*B1, *rpo*B2, *rpo*B3, *rpo*B4, *kat*G, *inh*A) can be checked by the operator. The sample is defined "RIF Resistance Negative" and "INH Resistance Negative" if all the Tm values fall within the limits reported in Table 2.

Samples with an invalid result, due to Internal Control failure (incorrect extraction or inhibitors carry-over), were re-tested starting from extracted DNA diluted 1:4 in water.

### MTBc clinical strain characterization and titration

MTBc strains (n = 4) were characterized for mutation in *inhA*, *katG* or *rpoB* genes by Geno-Type MTBDRplus VER. 2.0. The results are shown in Table 3.

To titre MTBc strains 1:10 serial dilutions of recent (no more than 2 days) positive MGIT of each MTBc isolate were prepared in 0.9% saline solution. 250 μL of each dilution were inoculated onto solid culture in order to obtain the number of CFU/mL.

### Samples spiked with mutated MTBc strains

20 sputa, 20 urine samples, 20 biopsies, 20 cavitary fluids and 20 gastric aspirates, previously decontaminated and found Xpert and culture-negative, were spiked with 5,000 CFU/ml of MTBc mutated strain 2. This was repeated for strains 3 and 4 giving a total of 60 MTBc-resistant sputa and 240 MTBc-resistant extra-pulmonary samples.

### Statistical analysis

Determination of sensitivity, specificity, their 95% confidence intervals and Receiver Operating Characteristic (ROC) curve were performed using GraphPad Prism version 8.0.1 (Prism 8, San Diego, CA),.

To assess the agreement between phenotypic resistance and molecular typing obtained with ELITe as well as agreement between Elite and Xpert MTB/Rif, Cohen's κ statistics were calculated using Prism 8.

## Results

### Sample inactivation

Heat-inactivated specimens were MTBc culture-negative after 8 weeks of incubation at 37°C, demonstrating the efficacy of heat inactivation by dry block at 95°C.

### Invalid results

Six (1.0%) of the 560 tests performed by ELITe were invalid: 2 (1.3%) of the 160 pulmonary samples and 4 (1.0%) of the 400 extra-pulmonary samples (2 biopsies, 1 cavitary liquid and 1 gastric aspirate). All samples with an invalid result were re-tested and a valid result obtained.

**Table 2. Melting temperature (Tm) limits for rifampicin and isoniazid susceptibility.**

| Probe | Tm Limits | Outcome |
|:---:|:---:|:---:|
| *rpo*B1 | $66.0 \leq Tm \leq 80.0$ | RIF Resistance Negative |
| *rpo*B2 | $70.0 \leq Tm \leq 80.0$ | |
| *rpo*B3 | $68.0 \leq Tm \leq 80.0$ | |
| *rpo*B4 | $63.5 \leq Tm \leq 80.0$ | |
| *kat*G | $69.0 \leq Tm \leq 80.0$ | INH Resistance Negative |
| *inh*A | $66.0 \leq Tm \leq 80.0$ | |

**Table 3. Molecular characterization of MTBc strains by GenoType MTBDRplus.**

| Name | *rpo*B | *inh*A | *kat*G |
|---|---|---|---|
| MTBc strain 1 | wt | wt | wt |
| MTBc strain 2 | mut3 | wt | wt |
| MTBc strain 3 | wt | mut1 | wt |
| MTBc strain 4 | wt | wt | mut1 |

## ELITe sensitivity and specificity

The overall sensitivity and specificity on both pulmonary and extra-pulmonary samples compared to culture was 90.77% (95% CI: 0.8456–0.9464) and 99.23% (95% CI: 0.9577–0.9996) respectively. ROC curve (Fig 1) shows the tradeoff between sensitivity and specificity, and the parameters were calculated for each Ct value. The area under the ROC curve was 0.95 (95% CI: 0.92–0.98), with a p value <0.0001.

## Performance on pulmonary samples

As shown in Table 4, 49 of the 50 MTBc culture-negative sputum samples tested by ELITe were found negative and 1 positive. 49 of the 50 MTBc culture-positive samples tested by ELITe were found positive and 1 negative. Therefore, the overall sensitivity and specificity of ELITe compared to culture were 98.0% [CI: 89.35–99.95] and 98.0% [CI: 89.35–99.95] respectively. Six of the 50 MTBc culture-positive sputa were smear-negative; 5 of these were ELITe positive.

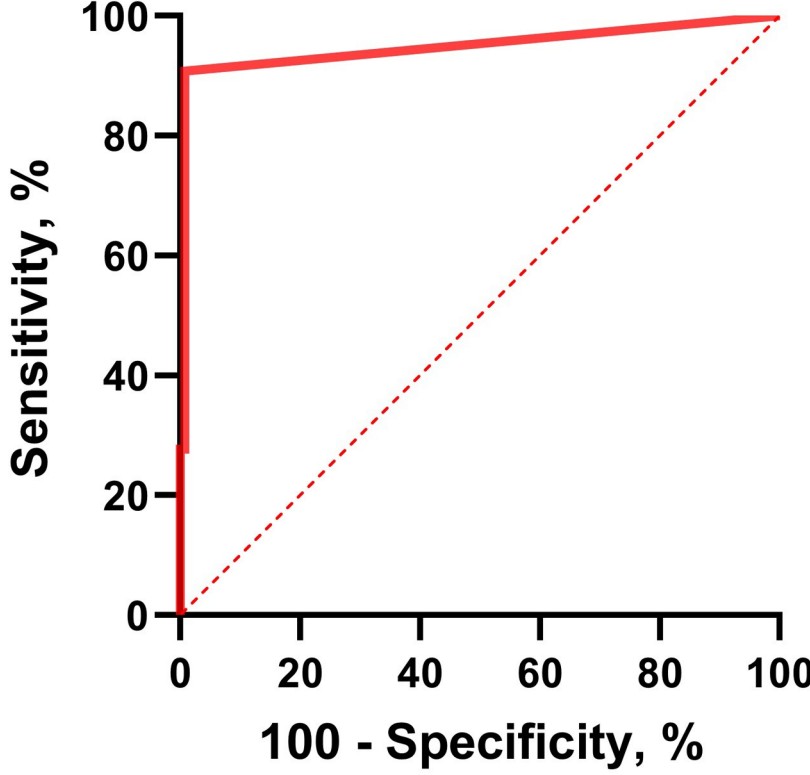

**Fig 1. ROC curve of the ELITe system for the diagnosis of tuberculosis.** TB diagnosis was assessed based on MTBc culture results.

**Table 4. Agreement between MDR/MTB Elite MGB® and MTBc culture.**

| MDR/MTB ELITe MGB® | SPUTUM (agreement %) | URINE (agreement %) | BIOPSY (agreement %) | CAVITARY FLUID (agreement %) | GASTRIC ASPIRATE (agreement %) |
|---|---|---|---|---|---|
| MTBc culture negative | 49/50 (98) | 20/20 (100) | 20/20 (100) | 20/20 (100) | 20/20 (100) |
| MTBc culture positive | 49/50 (98) | 16/20 (80) | 17/20 (85) | 19/20 (95) | 17/20 (85) |
| Spiked with MTBc strain 2 (*rpoB*) | 20/20 (100) | 20/20 (100) | 20/20 (100) | 20/20 (100) | 20/20 (100) |
| Spiked with MTBc strain 3 (*inhA*) | 20/20 (100) | 20/20 (100) | 20/20 (100) | 20/20 (100) | 20/20 (100) |
| Spiked with MTBc strain 4 (*katG*) | 20/20 (100) | 20/20 (100) | 20/20 (100) | 20/20 (100) | 20/20 (100) |

Molecular typing was feasible in 47 of the 49 drug susceptible MTBc-positive sputum samples with 100% agreement between genotypic susceptibility to Isoniazid and Rifampicin and phenotypic results. For these samples the mean ELITe MTBc Ct value was 24.2 ±4.0 (range: 18.13–34.11) and Time to Positivity in liquid culture (TTP) was 7.1 ± 3.0 days.

Molecular typing was not feasible in 2 of the 49 drug susceptible MTBc-positive samples: MTB Ct Values were 35.90 and 37.79 respectively, smear microscopy was negative and mean TTP was 14.5±3.5 days.

Sputum samples (n = 60) spiked with 3 mutated MTBc strains were found positive with 100% molecular agreement for the *inhA*, *katG* and *rpoB* genes (Table 4).

Table 5 shows molecular agreement between ELITe and Xpert for the 100 sputa tested with Xpert. 49 were Xpert-, ELITe- and culture-positive, 49 were Xpert-, ELITe- and culture-negative. Two samples gave discordant results (1 was Xpert and MTBc culture-positive/ELITe negative and 1 was Xpert and MTBc culture-negative /ELITe positive), with an overall agreement of 98% [K: 0.96, CI: 0.905–1].

## Performance on extra-pulmonary samples

As shown in Table 4, all (n = 80) MTBc culture-negative extra-pulmonary samples (urine, biopsy, cavitary fluid and gastric aspirate) were found negative by ELITe, with a specificity of 100% [CI: 95.49–100].

Excluding 8 spiked urine samples, in MTBc culture-positive samples (n = 72), the overall sensitivity of ELITe on extra-pulmonary samples was 86.3% [CI: 76.73–92.93]; sensitivity was 80% [CI: 56.34–94.27] for urine, 85% [CI: 62.11–96.79] for biopsy, 95% [CI: 75.13–99.87] for cavitary fluid and 85% [CI: 62.11–96.79] for gastric aspirate. In 62 smear-negative extra-pulmonary samples, the overall sensitivity of ELITe was 82.3% [CI: 70.47–90.80].

Molecular typing was feasible in the following samples: 3/12 urine primary samples (mean MTBc Ct Value: 31.1±4.1), 12/17 biopsy (mean MTBc Ct Value: 30.0±3.9), 13/19 cavitary fluid (mean MTBc Ct Value: 30.1±2.6) and 2/17 gastric aspirate (mean MTBc Ct Value: 30.6±3.1),

**Table 5. Molecular agreement between MDR/MTB ELITe MGB® Kit and Xpert MTB/Rif.**

| Sample types | n. of samples tested with Xpert | Agreement (%) | 95% C.I. | Cohen's kappa |
|---|---|---|---|---|
| Sputum | 100 | 98 | 90.5–100 | 0.960 |
| Urine | 30 | 90 | 45.2–100 | 0.734 |
| Biopsy | 40 | 92.5 | 68.9–100 | 0.850 |
| Cavitary Fluid | 40 | 97.5 | 85.3–100 | 0.950 |
| Gastric Aspirate | 40 | 92.5 | 68.2–100 | 0.848 |

with 100% agreement with the phenotypic test. In culture-positive samples where molecular typing was not feasible (n = 35), the mean MTB Ct Value was 38.0 ±2.0; these samples were smear-negative. TTPs were 14.3 ±3.2 days in samples where molecular typing was not feasible and 12.2 ±4.0 days in samples with feasible molecular typing. Extra-pulmonary samples (60 per matrix) spiked with 3 mutated MTBc strains were found positive with 100% molecular agreement for the *inhA*, *katG* and *rpoB* genes (Table 4).

The overall molecular agreement between ELITe and Xpert for extra-pulmonary samples was 93.33% [CI: 0.781–0.945, k = 0.863]; the agreement for each biological matrix is shown in Table 5. The best result was obtained with cavitary fluid (97.5%). Discordant results were due to 7 MTBc culture-positive samples being found Xpert positive/ELITe negative, and 3 MTBc culture-positive samples being found Xpert negative/ELITe positive.

## Discussion

This is the first assessment of MDR/MTB ELITe MGB® Kit (ELITe) performance on pulmonary and extra-pulmonary samples in comparison to MTBc culture in the literature to date.

The overall (pulmonary and extra-pulmonary) sensitivity and specificity compared to MTBc culture were 90.77% and 99.23% respectively, and ROC analysis (AUC = 0.95) showed the high accuracy of the ELITe system for the diagnosis of tuberculosis.

On sputum samples, both the sensitivity and specificity of ELITe were excellent (98% and 98% respectively); only two samples gave discordant results. One sputum was ELITe negative and culture-positive; in this case MTBc concentration was very low as shown by a long time to positivity (TTP) in liquid media (15.6 days). On the other hand, one sputum sample was ELITe positive but MTBc culture-negative. For this sample we can speculate a false positive result by ELITe or a true MTBc positive paucibacillary sample where culture isolation was not achieved; the high Ct value (Ct = 36) measured by ELITe could support the latter hypothesis, but unfortunately clinical data were not available.

One limit of this study is that most sputum samples were smear-positive, therefore sensitivity on smear-negative samples could not be evaluated. However, 5 of the 6 smear-negative samples were ELITe positive. In our setting most sputa from TB patients are smear-positive, while most smear-negative pulmonary samples are bronchoalveolar fluids, which were not included in this study.

For extra-pulmonary samples including urine, cavitary fluids, biopsy and gastric aspirates, ELITe specificity was 100% and sensitivity was 86.3%. Furthermore, sensitivity in smear-negative MTBc culture-positive extra-pulmonary samples was 82.3%. Most studies addressing the detection of MTBc by molecular tests on extra-pulmonary samples have been based on Xpert MTB/RIF (according to Tortoli *et al.* sensitivity: 81.3% [6]; Mazzola *et al.* sensitivity: 83.6% [11]; Lombardi *et al.* sensitivity: 76.8% [12]) or on Xpert Ultra systems (according to Perez-Risco *et al.* sensitivity in smear-negative e-PTB: 75.9% [13]; Opota *et al.* sensitivity: 83.7% [14]). Data obtained with ELITe are in line with results obtained with those systems.

Molecular typing to detect Isoniazid and Rifampicin resistance was feasible in 96% sputum and 46% extra-pulmonary culture-positive samples. In sputa where typing was not feasible, MTBc Ct values and TTP in liquid culture were higher than those in which typing was feasible (Ct values 36.8 vs. 24.2, TTP 14.5 vs. 7.1 days, respectively), indicating a low concentration of MTBc DNA. Similar results were obtained for extra-pulmonary samples.

Detection of Rifampicin and Isoniazid resistance by ELITe was assessed on 60 MTB spiked sputum samples and 240 MTB spiked extra-pulmonary samples, with a molecular agreement of 100% for the *inhA*, *katG* and *rpoB* genes. One limit of this study is that only 3 mutated MTBc strains were used, which do not cover the spectrum of possible mutations that confer

resistance to Isoniazid and Rifampicin. However, the entire spectrum of mutations was previously evaluated by the Company as described in the MDR/MTB ELITe MGB® Kit Instruction for Use [8].

Finally, the molecular agreement between ELITe and Xpert was excellent for sputum, cavitary fluid, biopsy and gastric aspirate, and good for urine, as shown by Cohen's Kappa. However, the sample volume used by the two assays is different (200 μL for ELITe, 500 μL for Xpert) and this difference could impact the assay comparison. In contrast to Xpert, ELITe identifies Isoniazid resistant strains, which would otherwise not be detected before phenotypic DST results or by a combination of Xpert and Line Probe Assay, such as GenoType MTBDRplus.

In this study, the overall rate of invalid results was very low (1.0%), with no difference between pulmonary and extra-pulmonary samples, confirming the robustness of the amplification process. Invalid results were lower than those obtained with Xpert (2.8%) described by Lombardi at al. [12].

This study was performed on frozen samples stored at -20˚C over a period of 18 months prior to assay. However, we can speculate that storage did not reduce specimen stability, as shown in our previous study with Xpert Ultra, where we demonstrated that the detection rate of the assay did not significantly decrease despite long-term storage of pulmonary and extra-pulmonary samples [15]. In order to perform the test in a Biosafety Level 2 laboratory, samples were previously inactivated. MTBc inactivation was achieved using a dry-block at 95˚C for 30 minutes, this allowed easier and safer heat-inactivation compared to a boiling water bath, as previously shown by other Authors [16, 17].

In conclusion, the high sensitivity and specificity results obtained in this study on a large sample size of specimens provide evidence for the use of MDR/MTB ELITe MGB® Kit in combination with ELITe InGenius® for the diagnosis of MTB complex as well as Rifampicin and Isoniazid resistance in both pulmonary and extra-pulmonary samples. This automated system simplifies the laboratory workflow, has a fast turnaround time (less than 3 hours) and is an aid in choosing appropriate therapeutic treatment and patient management.

## Acknowledgments

The authors thank ELITechGroup for providing MDR/MTB ELITe MGB® Kit reagents, Dr. Paola Monari and Dr. Eleonora Gatti for technical support and Jackie Leeder, BSc, for English language editing.

## Author Contributions

**Conceptualization:** Francesco Bisognin, Giulia Lombardi, Paola Dal Monte.

**Data curation:** Francesco Bisognin, Giulia Lombardi, Chiara Finelli, Paola Dal Monte.

**Funding acquisition:** Paola Dal Monte.

**Methodology:** Francesco Bisognin, Giulia Lombardi, Chiara Finelli, Paola Dal Monte.

**Project administration:** Maria Carla Re, Paola Dal Monte.

**Software:** Francesco Bisognin, Giulia Lombardi.

**Supervision:** Maria Carla Re.

**Validation:** Francesco Bisognin, Chiara Finelli.

**Visualization:** Maria Carla Re.

**Writing – original draft:** Francesco Bisognin, Giulia Lombardi, Paola Dal Monte.

**Writing – review & editing:** Francesco Bisognin, Giulia Lombardi, Paola Dal Monte.

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
