## [Decision Letter · Decision Letter 0]

31 Mar 2020

PONE-D-20-03632

Simultaneous detection of Mycobacterium tuberculosis complex and resistance to Rifampicin and Isoniazid by MDR/MTB ELITe MGB® Kit for the diagnosis of tuberculosis

PLOS ONE

Dear Dr. Lombardi,

Thank you for submitting your manuscript to PLOS ONE. After careful consideration, we feel that it has merit but does not fully meet PLOS ONE’s publication criteria as it currently stands. Therefore, we invite you to submit a revised version of the manuscript that addresses the points raised during the review process.

We would appreciate receiving your revised manuscript by May 15 2020 11:59PM. To enhance the reproducibility of your results, we recommend that if applicable you deposit your laboratory protocols in protocols.io, where a protocol can be assigned its own identifier (DOI) such that it can be cited independently in the future. For instructions see: http://journals.plos.org/plosone/s/submission-guidelines#loc-laboratory-protocols

We look forward to receiving your revised manuscript.

Kind regards,

HASNAIN SEYED EHTESHAM

Academic Editor

PLOS ONE

Additional Editor Comments (if provided):

Some classic publications on drug resistance published several years ago are missing in the references, the authors may need to include them. I recommend a minor revision.

Journal Requirements:

2. Thank you for including the following ethics statement for participant consent on the submission details page:

'Consent was not required as the data were analyzed anonymously.'

Please also include this consent information in the ethics statement in the Methods section of your manuscript.

Reviewers' comments:

Reviewer's Responses to Questions

**Comments to the Author**

1. Is the manuscript technically sound, and do the data support the conclusions?

Reviewer #1: Yes

Reviewer #2: Yes

2. Has the statistical analysis been performed appropriately and rigorously? 

Reviewer #1: No

Reviewer #2: Yes

3. Have the authors made all data underlying the findings in their manuscript fully available?

Reviewer #1: Yes

Reviewer #2: Yes

4. Is the manuscript presented in an intelligible fashion and written in standard English?

Reviewer #1: Yes

Reviewer #2: Yes

5. Review Comments to the Author

Reviewer #1: Comments to the authors:

The retrospective study titled “Simultaneous detection of Mycobacterium tuberculosis complex and resistance to Rifampicin and Isoniazid by MDR/MTB ELITe MGB® Kit for the diagnosis of tuberculosis” by Bisognin F et al. demonstrated the use PCR based MDR/MTB ELITe MGB® Kit in the detection of Mycobacterium tuberculosis complex (MTBc). Timely detection is required for efficient management of TB cases word-wide. The diagnosis methods which can be operated in resource limited settings are required for efficient management of tuberculosis or disease caused by the MTBc. In this retrospective study the authors used the stored samples isolated from pulmonary, and extra-pulmonary sites. They compared their results with AFB staining, culture as well as Xpert MTB/RIF. Although, they showed good agreement in specificity and sensitivity with all the known methods used for the detection of MTBc in pulmonary and extra-pulmonary samples. To demonstrate their results in statistically significant and more reliable way the authors should generate ROC curve for their findings and include in the manuscript. This will accurately demonstrate their findings in statistically significant and meaning full way. The manuscript is well written and organized. The conclusions drawn are well supported by their data. The authors should have included more number of cases of smear negative samples. The authors mentioned MTBc through the manuscript. They should introduce few sentences describing species included. The manuscript could be accepted for publication after careful revision and addition of ROC analysis.

Reviewer #2: Current manuscript is technically sound and evaluated in a planned way. Statistical analysis is done appropriately.

6. PLOS authors have the option to publish the peer review history of their article (what does this mean?). If published, this will include your full peer review and any attached files.

Reviewer #1: Yes: Mohd Shariq

Reviewer #2: No

---

## [Author Response · Author response to Decision Letter 0]

14 Apr 2020

Manuscript Re-submission to PLOS ONE: Response to Reviewers (PONE-D-20-03632)

Title: Simultaneous detection of Mycobacterium tuberculosis complex and resistance to Rifampicin and Isoniazid by MDR/MTB ELITe MGB® Kit for the diagnosis of tuberculosis 

Authors: Francesco Bisognin, Giulia Lombardi, Chiara Finelli, Maria Carla Re, Paola Dal Monte

Dear Editor,

we would like to submit the revised version of our manuscript (PONE-D-20-03632) in order to make it suitable for publication in PLOS ONE.

We would like to thank you and the Reviewers for the useful comments provided. Based on all comments received, we have revised the original manuscript and added the ROC curve required. 

The point-by-point responses to journal requirements and reviewers’ comments as well as the changes made to the original manuscript are described below.

Additional Editor Comments (if provided):

Some classic publications on drug resistance published several years ago are missing in the references, the authors may need to include them. I recommend a minor revision.

Thank you for your comment: we added 2 references about drug resistant MTBc in the Introduction section: 

• Rockwood N, Abdullahi LH, Wilkinson RJ, Meintjes G. Risk Factors for Acquired Rifamycin and Isoniazid Resistance: A Systematic Review and Meta-Analysis. PLoS ONE. 2015; 10(9):e 0139017.

• Albanna AS, Menzies D. Drug-resistant tuberculosis: what are the treatment options?. Drugs. 2011; 71(7):815-25. 

JOURNAL REQUIREMENTS:

Plos One style requirements have been satisfied.

2. Thank you for including the following ethics statement for participant consent on the submission details page:

'Consent was not required as the data were analyzed anonymously.'

Please also include this consent information in the ethics statement in the Methods section of your manuscript.

We included the sentence required about the Informed Consent in the Material and Method section (lines 94-95).

Here, we provide the DOI necessary to access our data.

Please update our Data Availability statement to: “All de-identified data used in this study are available from the Database Bisognin F et al ELITe, held in the Figshare public repository at the following URL: https://doi.org/10.6084/m9.figshare.12123585.v1.

Reviewers' comments:

Reviewer #1: Comments to the authors:

The retrospective study titled “Simultaneous detection of Mycobacterium tuberculosis complex and resistance to Rifampicin and Isoniazid by MDR/MTB ELITe MGB® Kit for the diagnosis of tuberculosis” by Bisognin F et al. demonstrated the use PCR based MDR/MTB ELITe MGB® Kit in the detection of Mycobacterium tuberculosis complex (MTBc). Timely detection is required for efficient management of TB cases word-wide. The diagnosis methods which can be operated in resource limited settings are required for efficient management of tuberculosis or disease caused by the MTBc. In this retrospective study the authors used the stored samples isolated from pulmonary, and extra-pulmonary sites. They compared their results with AFB staining, culture as well as Xpert MTB/RIF. Although, they showed good agreement in specificity and sensitivity with all the known methods used for the detection of MTBc in pulmonary and extra-pulmonary samples. To demonstrate their results in statistically significant and more reliable way the authors should generate ROC curve for their findings and include in the manuscript. This will accurately demonstrate their findings in statistically significant and meaning full way. The manuscript is well written and organized. The conclusions drawn are well supported by their data. The authors should have included more number of cases of smear negative samples. The authors mentioned MTBc through the manuscript. They should introduce few sentences describing species included. The manuscript could be accepted for publication after careful revision and addition of ROC analysis.

Many thanks for your comments.

As you suggested, we generated a ROC curve (Figure 1) to improve our statistical analysis. ROC analysis (AUC= 0.95) showed the high accuracy of the ELITe system for the diagnosis of tuberculosis.

Therefore we added the description of the statistical analysis used (in the Material and Method section), a paragraph in the Results section (lines 182-192), and a sentence in the Discussion section (lines 250-252).

Furthermore, as you suggested, we added a sentence describing MTB complex in the Introduction section (lines 49-51).

Reviewer #2: Current manuscript is technically sound and evaluated in a planned way. Statistical analysis is done appropriately.

Many thanks for your positive comments.

We hope that the revised manuscript can now be accepted for publication in PLOS ONE.

Thank you very much for your kind consideration.

Yours faithfully,

 Giulia Lombardi

Corresponding author: 

Dr. Giulia Lombardi, PhD

Department of Experimental, Diagnostic and Specialty Medicine - Microbiology Unit 

University of Bologna - S. Orsola Malpighi University Hospital 

Via Massarenti 9 - 40138 Bologna - Italy

---

## [Editor Report · Decision Letter 1]

20 Apr 2020

Simultaneous detection of Mycobacterium tuberculosis complex and resistance to Rifampicin and Isoniazid by MDR/MTB ELITe MGB® Kit for the diagnosis of tuberculosis

PONE-D-20-03632R1

Dear Dr. Lombardi,

We are pleased to inform you that your manuscript has been judged scientifically suitable for publication and will be formally accepted for publication once it complies with all outstanding technical requirements.

With kind regards,

HASNAIN SEYED EHTESHAM

Academic Editor

PLOS ONE

Additional Editor Comments (optional):

The authors have revised the manuscript. The only major issue related to statistical analyses and these have been addressed.
---

## [Editor Report · Acceptance letter]

23 Apr 2020

PONE-D-20-03632R1 

Simultaneous detection of *Mycobacterium tuberculosis* complex and resistance to Rifampicin and Isoniazid by MDR/MTB ELITe MGB^®^ Kit for the diagnosis of tuberculosis 

Dear Dr. Lombardi:

I am pleased to inform you that your manuscript has been deemed suitable for publication in PLOS ONE. Congratulations! Your manuscript is now with our production department. 

With kind regards,

on behalf of

Prof HASNAIN SEYED EHTESHAM 

Academic Editor

PLOS ONE